# *HFR1*, a bHLH Transcriptional Regulator from *Arabidopsis thaliana*, Improves Grain Yield, Shade and Osmotic Stress Tolerances in Common Wheat

**DOI:** 10.3390/ijms231912057

**Published:** 2022-10-10

**Authors:** Guanghua Sun, Luhao Yang, Weimin Zhan, Shizhan Chen, Meifang Song, Lijian Wang, Liangliang Jiang, Lin Guo, Ke Wang, Xingguo Ye, Mingyue Gou, Xu Zheng, Jianping Yang, Zehong Yan

**Affiliations:** 1Triticeae Research Institute, Sichuan Agricultural University, Chengdu 611130, China; 2State Key Laboratory of Wheat and Maize Crop Science, Center for Crop Genome Engineering, Longzi Lake Campus, College of Agronomy, Henan Agricultural University, Zhengzhou 450046, China; 3Institute of Radiation Technology, Beijing Academy of Science and Technology, Beijing 100875, China; 4Institute of Crop Sciences, Chinese Academy of Agricultural Sciences, Beijing 100081, China

**Keywords:** *HFR1*, grain yield, osmotic stress, shade avoidance, *Triticum aestivum*

## Abstract

Common wheat, *Triticum aestivum*, is the most widely grown staple crop worldwide. To catch up with the increasing global population and cope with the changing climate, it is valuable to breed wheat cultivars that are tolerant to abiotic or shade stresses for density farming. *Arabidopsis* LONG HYPOCOTYL IN FAR-RED 1 (AtHFR1), a photomorphogenesis-promoting factor, is involved in multiple light-related signaling pathways and inhibits seedling etiolation and shade avoidance. We report that overexpression of *AtHFR1* in wheat inhibits etiolation phenotypes under various light and shade conditions, leading to shortened plant height and increased spike number relative to non-transgenic plants in the field. Ectopic expression of *AtHFR1* in wheat increases the transcript levels of *TaCAB* and *TaCHS* as observed previously in *Arabidopsis*, indicating that the *AtHFR1* transgene can activate the light signal transduction pathway in wheat. *AtHFR1* transgenic seedlings significantly exhibit tolerance to osmotic stress during seed germination compared to non-transgenic wheat. The *AtHFR1* transgene represses transcription of *TaFT1*, *TaCO1*, and *TaCO2*, delaying development of the shoot apex and heading in wheat. Furthermore, the *AtHFR1* transgene in wheat inhibits transcript levels of *PHYTOCHROME-INTERACTING FACTOR 3-LIKEs* (*TaPIL13*, *TaPIL15-1B*, and *TaPIL15-1D*), downregulating the target gene *STAYGREEN* (*TaSGR*), and thus delaying dark-induced leaf senescence. In the field, grain yields of three *AtHFR1* transgenic lines were 18.2–48.1% higher than those of non-transgenic wheat. In summary, genetic modification of light signaling pathways using a photomorphogenesis-promoting factor has positive effects on grain yield due to changes in plant architecture and resource allocation and enhances tolerances to osmotic stress and shade avoidance response.

## 1. Introduction

Common wheat (*Triticum aestivum* L.) is the most widely grown staple crop worldwide, with an annual planting area of 215,901,958 hectares (http://www.fao.org/faostat/en/#data/QCL (accessed on 22 December 2021)). It is estimated that by 2050, the global population will grow by 25%, reaching 10 billion [1]. With the rapid increase in global population, demand for wheat is quickly increasing. In addition, extreme weather conditions are becoming increasingly frequent [2,3,4]. With the intensification of extreme climatic conditions, the resistance of wheat to abiotic stresses must be greatly enhanced to meet the challenge of feeding a population of 10 billion [5]. Another key point in crop breeding is increasing lodging resistance, which increases grain yields [5]. Therefore, the mechanism of wheat resistance to abiotic stress is worth exploring through biotechnology with the goal of wheat improvement [6,7,8].

Plants rely on a series of photoreceptors (such as phytochromes, phy; cryptochromes, cry; phototropins, phot; and UVR8) and intermediate factors to monitor environmental light cues and modulate their developmental processes, including seed germination, seedling de-etiolation, flowering time, and biomass production, crop quality, and yield [9,10,11,12,13,14,15,16,17,18]. Downstream of the light signaling pathway, COP1 (CONSTITUTIVE PHOTOMORPHOGENIC 1)-SPA1 (SUPPRESSOR OF PHYTOCHROME A-105) E3 complex mediates the degradation of a group of photomorphogenesis-promoting factors, including ELONGATED HYPOCOTYL5 (HY5), LONG HYPOCOTYL IN FAR-RED1 (HFR1), LONG AFTER FAR-RED LIGHT1 (LAF1), PHYTOCHROME RAPIDLY REGULATED 1 (PAR1), phyA, and phyB, to promote seedling etiolation and shade response [19,20,21,22,23,24,25,26,27]. To balance HY5, HFR1, and other positive factors, *PHYTOCHROME INTERACTING FACTORS* (*PIFs*) and *PHYTOCHROME-INTERACTING FACTOR 3-LIKEs* (*PILs*), which are members of the basic helix-loop-helix (bHLH) transcription factor family, function as central regulatory components in multiple signaling pathways [28,29,30,31].

*Arabidopsis HFR1* (*AtHFR1*) was originally defined as a positive factor in the phyA pathway mediating far-red (FR) light signaling [32,33,34]. Subsequently, involvement of AtHFR1 has been confirmed in the blue [35,36], red [37], and ultraviolet light [9,38] signaling pathways, which mediates seedling de-etiolation [23,24,26], shade avoidance [28,39], seed germination [40,41], floral induction [42,43], leaf senescence [44,45], and thermomorphogenesis [46,47]. As a bHLH transcriptional regulator, HFR1 participates in the regulation of plant development by interacting with PIFs and other proteins rather than directly binding to DNA [9,34,41]. When subjected to shading by vegetation, plants rapidly modulate a suite of developmental responses, referred to as shade avoidance syndrome (SAS), including a rapid elongation of stems and leaves, reduced leaf chlorophyll content, and increased apical dominance (early flowering and reduced branching) [48,49,50]. Compared with the wild type, the *AtHFR1*-deficient mutant displays more slender and early flowering in canopy shade, while overexpression of *AtHFR1* leads to SAS inhibition [22,24,39]. AtHFR1 is capable of binding to both CONSTANS (CO) and PIF7 proteins, blocking them from binding to the promoters of *FLOWERING LOCUS* (*FT*) and *pri-MIR156E/F*, respectively, which results in the inhibition of early flowering [42]. AtHFR1 also blocks PIF1 transcriptional activity through the formation of a heterodimer with PIF1 to promote seed germination [40,41].

The light signal transduction pathway is closely associated with plant growth and development. Increasing or reducing components in the light signaling pathway can lead to significant changes in the development of plants [11,51,52,53]. Similar to the *phyB* mutant of *Arabidopsis*, spontaneous *phyB* mutants of sorghum (*Ma3*) [54], barley (*BMDR-1*) [55], rice (*se5*) [56], and maize (*elm1*) [57] exhibit early flowering, while the *phyC* mutation in wheat leads to significant heading delay [58]. Transgenic potato plants expressing *Arabidopsis PHYB* exhibit enhanced photosynthesis, a reduced shade-avoidance response, and higher tuber yields [52,53]. Overexpression of the *Arabidopsis PHYA* gene in rice reduces plant height and improves grain yield [51]. Several bHLH transcription factors from maize (*Zea mays*), rice (*Oryza sativa*), tomato (*Solanum lycopersicum*), and grape (*Vitis vinifera*) enhance tolerance to abiotic stress, plant cell size, biomass, and yield [59,60,61,62].

In the present study, we report that the ectopic expression of *AtHFR1* increases transcript levels of *chlorophyll a/b binding protein* (*CAB*) and *CHALCONE SYNTHASE* (*CHS*) in wheat plants as observed previously in *Arabidopsis*, indicating that the *AtHFR1* transgene can activate the light signal transduction pathway in wheat. Overexpression of *AtHFR1* in wheat inhibits etiolation phenotypes under various light and shade conditions, leading to shortened plant height, delayed heading, and increased spike number in the field relative to non-transgenic seedlings. The *AtHFR1* transgene also significantly represses the transcription of *TaFT1*, *TaCO1*, and *TaCO2*, delaying the development of shoot apex and heading in wheat. Furthermore, the *AtHFR1* transgene in wheat represses transcript levels of *TaPIL13*, *TaPIL15-1B*, and *TaPIL15-1D*, downregulates their target gene *STAYGREEN* (*SGR*), and thereby delays dark-induced leaf senescence. In addition, *AtHFR1* transgenic wheat seedlings show improved tolerance to osmotic stress during seed germination. In the field, grain yields of three *AtHFR1* transgenic lines were 18.2–48.1% higher than those of the WT wheat. These results indicate that the modification of light signaling pathways through ectopic expression of *AtHFR1* not only effectively improves tolerance to shade and abiotic stresses but also increases grain yield in wheat. Thus, genetic modification of light signaling pathways via a photomorphogenesis-promoting factor has positive effects on grain yield through changes in plant architecture and resource allocation.

## 2. Results

### 2.1. The AtHFR1 Transgene Represses Seedling Etiolation and Shade Avoidance in Common Wheat

Overexpression of *AtHFR1* in *Arabidopsis* promotes seedling de-etiolation and reduces shade avoidance syndrome (SAS) [23,24,26,28,39]. To explore the application of *AtHFR1* to wheat improvement, we generated *pBCXUN-AtHFR1* under the control of the ubiquitin promoter (Appendix A). This construct was introduced into the spring type cultivar Fielder (wild type, WT), and homozygous T_4_ transgenic lines of three high expression transgenic lines, designated #498, #511, and #515, were used for subsequent tests (Appendix A). Relative expression levels of the transgenic *AtHFR1* in Line #511 and #515 were 1.12- and 1.26-fold higher than that in Line #498 (Appendix A). These three transgenic lines were grown in darkness (Dk), far-red (FR), red (R), blue (B), or white (W) light conditions for 7 days to evaluate coleoptile elongation. Coleoptiles of the WT seedlings grown under continuous FR, R, B, or W light conditions were 78.2%, 67.6%, 65.6%, and 56.7% shorter than coleoptiles of seedlings grown in darkness, respectively (Figure 1A and Appendix A). Coleoptiles of the three transgenic lines reached only 70.7–84.3% of the length of the WT plants under Dk, FR, R, B, and W light conditions (Figure 1A and Appendix A). To provide further molecular evidence of the involvement of *AtHFR1* in wheat light signaling, we measured transcript levels of *chlorophyll a/b binding protein* (*CAB*) and *CHALCONE SYNTHASE* (*CHS*) in *AtHFR1* transgenic lines. Aside from Line #498 in blue light, all *TaCAB* transcript levels were significantly higher in *AtHFR1* transgenic lines than in the WT, especially under R light (Figure 1B). Except when treated with FR light, the two *AtHFR1* transgenic lines tested had much higher *TaCHS* transcript levels than the WT, particularly Line #511 in response to R light (Figure 1C).

Next, we tested the shade responses of two *AtHFR1* transgenic lines using both low-R/FR and low-B treatments. Under the high-R/FR condition, the coleoptile lengths of Lines #498 and #511 were 82.6% and 81.1% of that of the WT, while in the low-R/FR condition, the coleoptiles of Lines #498 and #511 were 77.6% and 77.8% of the length in the WT, respectively (Figure 1D). The lengths of the first true leaves of Lines #498 and #511 showed the same trend as their coleoptile lengths under both high-R/FR and low-R/FR conditions (Figure 1E). Under high-B conditions, the coleoptile lengths of Lines #498 and #511 were 77.6% and 75.6% of that of the WT, while under low-B conditions, the coleoptiles of Lines #498 and #511 were 81.6% and 80.2% of the length in the WT, respectively (Figure 1F). The lengths of the first true leaves of transgenic lines also showed a shortened phenotype under both high-B and low-B conditions. Thus, the increase in light responsiveness observed in *AtHFR1* transgenic lines activated light signal transduction and inhibited shade avoidance in wheat.

### 2.2. The AtHFR1 Transgene Delays Heading in Common Wheat

To identify the function of *AtHFR1* in wheat heading, we assessed the heading date and development of the shoot apical meristem in *AtHFR1* transgenic lines. We observed that the transgenic lines #498, #511, and #515 underwent heading 7.4, 5.3, and 4.8 days later than the WT, respectively (Figure 2A,B). Moreover, we observed slower development of the shoot apical meristem and inflorescence in Lines #498 and #511 compared to the WT (Figure 2C). The flowering locus T (*FT*) and *CO* genes play key roles in the control of flowering timing in higher plants, and therefore we examined *FT* and *CO* expression in the three transgenic lines and the WT. The *TaFT1*, *TaCO1*, and *TaCO2* genes showed lower expression levels under long-day conditions in transgenic plants than in the WT (Figure 2D–F and Appendix A). Based on these findings, ectopic expression of *AtHFR1* in cv. Fielder can significantly delay heading.

### 2.3. The AtHFR1 Transgene Leads to Compact Plant Architecture in the Field

Compared with non-transgenic wheat plants, adult plants of Lines #498, #511, and #515 were 79.8%, 81.6%, and 81.5% shorter in the field, respectively (Figure 3A,B; Appendix A). This difference was mainly driven by a significant reduction in the first three internode distances from the top in the transgenic lines. The average lengths of the first three internodes in the three transgenic lines were 69.1%, 83.0%, and 85.4% shorter than the corresponding lengths in the WT plants, respectively (Figure 3C,D; Appendix A). Strong changes caused by the *AtHFR1* transgene were observed in the angles of the tiller and flag leaf. Tiller angles of transgenic lines #498, #511, and #515 were 60.1%, 57.1%, and 58.1% smaller than the angles in non-transgenic seedlings, respectively (Figure 3E,F). The flag-leaf angles of transgenic lines #498, #511, and #515 were 61.8%, 53.0%, and 56.1% smaller than the angles in the WT, respectively (Figure 3G). The average length and width of flag leaves in transgenic lines were 68.0% and 74.5% smaller than the flag leaves of the WT (Figure 3H–I; Appendix A), while the average thickness and chlorophyll content of flag leaves in the three transgenic lines were 12.3% and 19.9% greater than in the WT (Figure 3J–L). Together, these results demonstrate that overexpression of *AtHFR1* alters adult morphology and leads to compact plant architecture and favorable leaf type related to the photosynthesis of wheat in the field.

### 2.4. The AtHFR1 Transgene Enhances Grain Yield per Plant in Wheat

After demonstrating that the *AtHFR1* transgene results in strong growth of wheat in the field, we assessed whether this gene can increase wheat yield. The number of spikes per plant of the WT was 33.3, and the average spike numbers per plant of the three transgenic lines were approximately 1.9 times higher than that of the WT (Figure 4A). No significant difference was observed between the three transgenic lines and the WT in grain number per spike (Figure 4B). The seed size (Appendix A) and weight (Figure 4C) of the three transgenic lines were somewhat smaller than that of the WT. Average grain width, length, and weight of the three transgenic lines were 94.9%, 91.8%, and 82.6% of the corresponding measurements in the WT (Appendix A and Figure 4C). Finally, the grain yields per plant of Lines #498, #511, and #515 were 18.2%, 38.0%, and 48.1% higher than the yield of the WT, respectively (Figure 4D). Although the *AtHFR1* transgenic lines had a smaller grain size than the WT, they still had higher grain yield due to their higher number of spikes per plant.

### 2.5. The AtHFR1 Transgene Enhances Wheat Tolerance to Osmotic Stress during Seed Germination

Previous research found that *AtHFR1* is involved in light-initiated seed germination [40,41]. To determine whether *AtHFR1* can improve the tolerance of wheat to osmotic stress during germination, we evaluated the seed germination rate and germination potential. Under mock (water) treatment, seed germination in the WT showed little lag compared with the *AtHFR1* transgenic lines. Seed germination rates of both non-transgenic and transgenic lines were over 96.7% 6 days after the mock treatment (Figure 5A). Seed germination began for the two transgenic lines on the fourth day with 150 mM NaCl treatment, which was 1 day earlier than in the mock treatment (Figure 5B). Large differences in the seed germination rate were found between non-transgenic plants and the *AtHFR1* transgenic lines with NaCl treatment (Figure 5B). On the tenth day after NaCl treatment, the seed germination rates of the two transgenic lines were over 80%, approximately 1.8 times the rate in non-transgenic plants (Figure 5B). Treatment with 20% polyethylene glycol (PEG) seriously delayed the germination of non-transgenic plants relative to the two transgenic lines (Figure 5C). On the tenth day after PEG treatment, the seed germination rates of the two transgenic lines were over 94%, or about 5.6 times the rates of the WT (Figure 5B). Next, we evaluated seed germination potential through the measurement of lengths of first true leaves under osmotic stress conditions. With mock (water) treatment, the average length of the first true leaf in the two transgenic lines reached only 86.2% of the length in non-transgenic plants (Figure 5D,E). After NaCl or PEG treatment for 8 days, the average length in the two transgenic lines was over 1.5 times that of non-transgenic plants (Figure 5D,E). Based on these results, the *AtHFR1* transgene enhances wheat tolerance to osmotic stress during seed germination.

### 2.6. The AtHFR1 Transgene Inhibits Dark-Induced Senescence in Wheat

Darkness is a common environmental factor that induces plant senescence [63]. To determine whether *AtHFR1* increases wheat tolerance to light starvation, we tested dark-induced senescence in the transgenic lines. After wheat seedlings were grown under long-day conditions (LD, 16 h light/8 h dark) for 8 days, they were transferred into Dk for 3, 6, 9, or 12 days (Figure 6A), or kept in LD for 12 days (Appendix A). After transfer into Dk for 6 days, leaf tops of the WT turned yellow, while those of the *AtHFR1* transgenic lines (#498 and #511) remained green (Figure 6A). After transfer into Dk for 12 days, half of the blades of the WT turned yellow and necrotic, while the leaves of transgenic lines (#498 and #511) had yellowing symptoms on only about one-third of the leaf tips (Figure 6A). After transfer into Dk for 12 days, the chlorophyll content was about 1.4 times higher in the transgenic lines than that in the WT (Figure 6B). These data suggest that the *AtHFR1* transgene delayed dark-induced leaf senescence in wheat.

*PIFs* and *PILs* target *STAYGREEN* (*SGR*) to induce leaf senescence during darkness treatment [44,45,63,64]. Although transcript levels of *SGR* in both transgenic lines and the WT increased significantly with light starvation treatment, *SGR* levels were much higher in the WT than that in the transgenic line (#498) (Figure 6C). The relative expression levels of *TaPIL13*, *TaPIL15-1B*, and *TaPIL15-1D* in the transgenic lines (#498) were lower than in the WT during dark treatment (Figure 6D–F). The above results suggest that the *AtHFR1* transgene inhibited dark-induced leaf senescence by reducing transcript levels of *TaPILs*.

## 3. Discussion

### 3.1. Overexpression of AtHFR1 Alters Plant Architecture by Activating the Light Signal Transduction Pathway in Wheat

Three lines of common wheat overexpressing the *Arabidopsis HFR1* gene were characterized (#498, #511, and #515) and were found to exhibit reduced seedling etiolation under FR, R, B, W, and even Dk conditions relative to non-transgenic plants, which is consistent with the results in *Arabidopsis* (Figure 1A) [23,24,26]. Overexpression of *AtHFR1* in wheat alters the adult plant architecture and supports strong growth in the field, with traits such as reduced plant height, shortened internode spacing, increased spike number per plant, decreased tiller angle and flag-leaf angle, and shortened and narrowed flag leaves with increased thickness (Figure 2A and Figure 4). Changes in adult architecture through a reduction in plant height and increase in branch number have been reported with ectopic expression of the oat *PHYA* gene in tobacco [65], *Arabidopsis PHYA* gene in rice [51], *Arabidopsis PHYB* gene in potato [52], and *PHYB* gene of Chinese cabbage in *Arabidopsis* [66]. Members of the CAB protein family are involved in pigment biosynthesis and assembly of the thylakoid membrane, and are considered early light-inducible proteins [67]. CHS is a ubiquitous enzyme in higher plants, and some CHS proteins are constitutively transcriptionally induced by changes in environmental factors such as light [23,68]. The ectopic *AtHFR1* transgene increased transcript levels of *TaCAB* and *TaCHS* in wheat plants, as reported previously in *Arabidopsis* (Figure 1B,C) [23]. Heading delay and activation of *TaFT1*, *TaCO1*, and *TaCO2* transcription with ectopic expression of *AtHFR1* in cv. Fielder is in accordance with results obtained in Arabidopsis (Figure 2) [42,43]. In addition, the *AtHFR1* transgene repressed transcript levels of *TaPILs* and *TaSGR* during dark-induced senescence (Figure 6C–F). Together, these results suggest that the *AtHFR1* transgene activates the light signal transduction pathway in common wheat.

### 3.2. Overexpression of AtHFR1 Effectively Improves Tolerance to Shade and Osmotic Stress in Wheat

Plants can detect a low-R/FR ratio through phytochromes or low-B light through cryptochromes [48,49,69]. Under low-R/FR or low-B conditions, the *AtHFR1* transgene represses elongation of both the coleoptile and the first true leaf (Figure 1D–G). Compared with non-transgenic wheat plants, *AtHFR1* transgenic seedlings display inhibition of SAS under field conditions, characterized by short plant height, delayed heading, increased spike number, narrowed and thickened leaves, and increased leaf chlorophyll content (Figure 2 and Figure 3). In wheat, the *AtHFR1* transgene inhibits transcript levels of *TaPIL13*, *TaPIL15-1B*, and *TaPIL15-1D*, thereby downregulating their target gene *TaSGR* and delaying dark-induced leaf senescence (Figure 6). Thus, the *AtHFR1* transgene effectively improves shade tolerance in wheat. In the *Arabidopsis* SAS, faster growth on the lower side than the upper side causes upward bending of leaves, called hyponasty [49]. Unexpectedly, the *AtHFR1* transgene decreases both tiller and flag-leaf angle relative to non-transgenic plants in the field (Figure 3F,G). In China, breeders have proposed that a large population (≥7000 thousand spikes/hm^2^) with a narrow, erect leaf type may be a suitable model for super wheat breeding [70]. An increased spike number per plant and an erectophile leaf type caused by the *AtHFR1* transgene are consistent with this super wheat model.

The relationship between the light signal transduction pathway and abiotic stress has been verified in many studies. PhyB enhances the accumulation of abscisic acid, thereby improving the tolerance of *Arabidopsis* to drought stress [71]. Photoactivated soybean GmCRY2a inhibits the expression of the senescence-related gene *WRKY DNA BINDING PROTEIN53b* (*WRKY53*) through direct interaction with CRYPTOCHROME-INTERACTING bHLH1 (CIB1), thereby inhibiting leaf senescence [72]. Tomato (*Solanum lycopersicum* L.) PIF4 (SlPIF4) directly activates expression of *SlDELLA* genes, promoting cold tolerance in tomato [61]. As an important transcriptional regulator, HFR1 is involved in numerous signaling pathways that regulate plant growth and development processes, including tolerance to abiotic stress [44,45,46,47]. Our results demonstrate that wheat seeds carrying the *AtHFR1* transgene enhance germination ability and germination potential relative to non-transgenic seeds under osmotic stress conditions (150 mM NaCl or 20% PEG) (Figure 5). Thus, the role of *HFR1* in the improvement of crop stress resistance merits further study.

### 3.3. Genetic Modification of the Light Signaling Pathways Has Positive Effects on Grain Yield through Changes in Resource Allocation

The peak period of traditional breeding for high yields was during the Green Revolution in the 1960s, which involved the widespread adoption of gibberellin (GA)-deficient semi-dwarf varieties to increase lodging resistance and support high nitrogen usage [5,51,73,74]. Dwarfism increases resistance to lodging and allows plants to support high grain yields; thus, the primary difficulty in developing high-yield cultivars is still related to lodging, even after the Green Revolution [5]. Because the genes introduced during the Green Revolution have negative effects on crop productivity due to the limitations of nitrogen-use efficiency [75], novel approaches to improving lodging resistance must be adopted [5]. Compared with non-transgenic seedlings, transgenic *AtHFR1* wheat plants exhibit not only shorter plant height and better plant architecture (Figure 3) but also improved grain yield and tolerance to shade, which may be helpful in dealing with wheat lodging (Figure 1, Figure 3, Figure 4 and Figure 6). The *AtHFR1* transgene also enhances the germination rate and germination potential of wheat seeds under osmotic stress (Figure 5). Another feature of transgenic *AtHFR1* wheat plants is that they display inhibited SAS at a low-R/FR ratio, as well as under low-B and in the field conditions (Figure 1 and Figure 3). Together, these results demonstrate that modification of the light signal transduction pathway in wheat via *AtHFR1* can effectively regulate plant architecture and subsequently improve grain yield and resistance to shade and osmotic stress.

## 4. Materials and Methods

### 4.1. Plasmid Construction and Genetic Transformation

Total RNA was extracted from Col-0 *Arabidopsis* seedlings grown in the dark using the Trizol reagent (Invitrogen, Waltham, MA, USA) and converted into first-strand cDNA using the RevertAid First Strand cDNA Synthesis Kit (Thermo Fisher Scientific, Waltham, MA, USA). The full-length coding region of *AtHFR1* (At1g02340) was isolated via PCR using the primers AtHFR1-F1 (5′-ATGTCGAATAATCAAGCTTTC-3′) and AtHFR1-R1 (5′-TCATAGTCTTCTCATCGCATG-3′), and then inserted into the *pBCXUN* vector digested with *Xcm* I to produce the *pBCXUN-Ubi::AtHFR1* (*pBCXUN-AtHFR1*) construct. The recombinant plasmid was sequenced at the Beijing AuGCT DNA-SYN Biotechnology Co., Ltd. (Beijing, China). The *pBCXUN-AtHFR1* binary construct was electroporated into *Agrobacterium tumefaciens* (strain *EHA105*) and then delivered to wheat cultivar Fielder (spring type) following the protocol described by Wang et al. [76]. Basta resistance evaluation and PCR were performed to appraise *AtHFR1* transgenic lines. After multi-generations of tests, homozygous transgenic lines were selected and used in subsequent experiments.

### 4.2. Plant Growth Conditions

Wheat seeds were planted in water-soaked vermiculite after being soaked for 12 h, and then grown under various light conditions. The plants used for coleoptile measurement were grown under various light conditions at 22 °C for 7 days. FR, R, and B light were supplied by light-emitting diode (LED) light sources (model 740FLED-2D; Taiwan Hipoint Co., Kaohsiung, China) at irradiance fluence rates of approximately 10, 15, and 15 µmol·m^−2^·s^−1^, respectively, unless otherwise indicated. White light was supplied using cool-white fluorescent lamps (100 µmol·m^−2^·s^−1^, unless otherwise indicated). Light fluence rates were measured using a model HR-350 spectrometer (Taiwan Hipoint Co., Kaohsiung, China). To analyze expression levels of the *TaCAB* and *TaCHS* genes, seedlings were grown in the dark at 22 °C for 7 days, and then transferred to the appropriate light conditions at 22 °C for 24 h. To evaluate the shade response, seedlings were grown under high-R/FR (R, 96 μmol·m^−2^·s^−1^; FR, 21 μmol·m^−2^·s^−1^; B, 15 μmol·m^−2^·s^−1^), low-R/FR (R 12 μmol·m^−2^·s^−1^; FR 105 μmol·m^−2^·s^−1^; B 15 μmol·m^−2^·s^−1^) [39], or high-B (50 μmol·m^−2^·s^−1^) or low-B (2 μmol·m^−2^·s^−1^) conditions at 22 °C for 7 days, and then the lengths of the coleoptiles and first true leaves were measured.

To observe the development of the main shoot apex, seedlings were grown under long-day conditions for 23, 42, or 52 DAG (days after germination). Long-day conditions entail 14 h light and 10 h dark at 22 °C with W light at 100 μmol·m^−2^·s^−1^. For light starvation, seedlings were grown under long-day conditions for 8 days, and then transferred to the dark for 3, 6, 9, or 12 days.

To test resistance to osmotic stress, wheat seeds were placed on filter paper soaked with water, 150 mM NaCl solution, or 20% PEG solution in Petri dishes, and then grown at 22 °C. Seed germination rates were investigated daily. Lengths of the coleoptile and first true leaf were measured on the 8th day.

### 4.3. Agronomic Traits

Planting and management of materials in the field (near Zhengzhou, ≈34.9° N, ≈113.6° E; sandy loam soil; annual rainfall 632 mm) followed methods used in general field production except row spacing and spacing in the rows were 60 × 20 cm to guarantee the plants grow well from October 2020 to May 2021. The date on which the main-stem spike emerged from the flag-leaf sheath was recorded as the heading date. After heading, the flag-leaf angle and tiller angle were investigated. The angle between the flag leaf and stem (upper) was recorded as the flag-leaf angle, and the angle between the tiller and the vertical (upper) was recorded as the tiller angle. Flag leaves were measured in terms of length, width, and thickness. The length between the pulvinus and tip was recorded as the leaf length, and the width at the middle of the leaf was measured as the leaf width. The thickness of the flag leaf at the filling stage was measured through slice analysis. After harvest, plant height, yield per plant, spike numbers per plant, grain number per spike, thousand-grain weight, grain length, and grain width were measured.

### 4.4. PCR and Real-Time Reverse Transcription PCR

Total RNA was extracted using the Eastep^®^ Super Total RNA Extraction Kit (Promega, Madison, WI, USA, LS1040) and the manufacturer’s instructions were followed. First-strand cDNA was synthesized using the GoScript™ Reverse Transcription Mix and Oligo (dT) (Promega, A2790). Wheat leaves at the jointing stage were used to extract DNA for identifying *AtHFR1* transgenic lines. The PCR reaction mixture consisted of 10 μL of 2 × Taq PCR Mix (Tiangen, Beijing, China, KT201), 0.5 μL each of the forward and reverse primers, 1 μL of DNA template, and 8 μL of H_2_O. An Applied Biosystems 2720 thermal cycler was used to perform PCR following the procedure of 94 °C for 5 min and 30 cycles of 94 °C for 20 s, 58 °C for 20 s, and 72 °C for 30 s. The relative expression levels (gray value) of *AtHFR1* were measured by ImageJ software (http://rsb.info.nih.gov/ij/download.html (accessed on 20 August 2021)).

Wheat leaves at the jointing stage or seedlings grown under adverse light treatments were used for real-time reverse transcription (RT)-PCR. The quantitative fluorescence reaction mixture consisted of 10 μL of 2× *PerfectStart*^®^ Green qPCR SuperMix (Transgen, Beijing, China, AQ601), 0.5 μL each of the forward and reverse primers, 2 μL of cDNA template, and 7 μL of H_2_O. A Roche LightCycler^®^ 480 II instrument was used to perform the fluorescence quantitative reaction with a profile of 94 °C for 30 s followed by 45 cycles of 94 °C for 5 s, 55 °C for 15 s, and 72 °C for 10 s. The change in threshold crossing (Ct) was used to calculate the relative expression level of each mRNA using the formula 2^−ΔΔCt^ [66]. Each reaction was performed in triplicate and relative expression levels were determined after normalization to the reference gene *TaACTIN* (AB181991.1). All the primers used above are listed in Appendix A.

### 4.5. Chlorophyll Extraction

Chlorophyll measurement was conducted according to the methods described previously [72,77].

### 4.6. Statistical Analysis

Significant differences in the data from this study were determined according to the results of variance (ANOVA) testing (ns indicates no significant difference; * *p* < 0.05; ** *p* < 0.01), and the WT (cv. Fielder) was used as the control in each test.

### 4.7. Accession Numbers

Sequence data from this article can be found in the *Arabidopsis* Genome Initiative or GenBank/EMBL databases under the following accession numbers: *AtHFR1* (At1g02340), *TaACTIN* (AB181991.1), *TaCAB* (XM_044587543.1), *TaCHS* (XM_044598705.1, XM_044526325.1), *TaFT1* (DQ890162.1) [78], *TaCO1* (XM_044588417.1, XM_044577719.1, XM_044569893.1) *TaCO2* (XM_044550689.1, XM_044558020.1, XM_044564766.1), *TaSGR* (XM_044526410.1, XM_044533918.1, XM_044542480.1), *TaPIL13* (XM_044530914.1, XM_044530915.1), *TaPIL15-1B* (XM_044531643.1, XM_044531644.1), *TaPIL15-1D* (XM_044592408.1, XM_044592409.1).

## Figures and Tables

**Figure 1 ijms-23-12057-f001:**
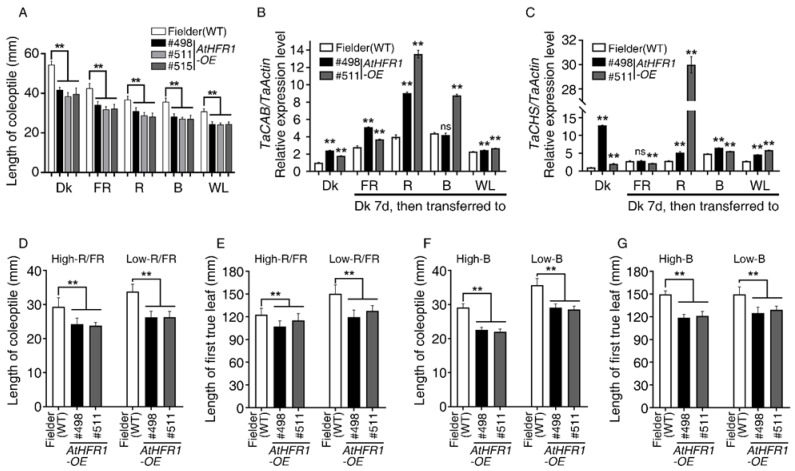
*AtHFR1* transgenic seedlings display promotion of de-etiolation under far-red (FR), red (R), blue (B), and white (W) light conditions and inhibition of shade avoidance under low-R/FR and low-B light. The fluence rates FR, R, B, and W light were 10, 15, 15, and 100 μmol·m^−2^·s^−1^, respectively, unless otherwise indicated: (**A**) Quantification of coleoptile lengths of the WT (cv. Fielder) and *AtHFR1* transgenic lines (#498, #511, and #515) (average of at least 30 seedlings) under various light conditions. Seedlings were grown under dark (Dk), FR, R, B, or W light conditions at 22 °C for 7 days. Data are means ± standard deviations (SDs). Real-time RT-PCR analysis of *TaCAB* (**B**) and *TaCHS* (**C**) in the WT and *AtHFR1* transgenic lines (#498 and #511). Seedlings were grown in darkness for 7 days and then transferred to the appropriate light conditions for 24 h. Data are means ± SDs, *n* = 3. Coleoptile length (**D**) and first true leaf length (**E**) of the WT and *AtHFR1* transgenic lines (#498 and #511) under high-R/FR (R, 96 μmol·m^−2^·s^−1^; FR, 21 μmol·m^−2^·s^−1^; B, 15 μmol·m^−2^·s^−1^) or low-R/FR (R 12 μmol·m^−2^·s^−1^; FR 105 μmol·m^−2^·s^−1^; B 15 μmol·m^−2^·s^−1^) conditions for 7 days. Data are means ± SDs, *n* ≥ 45. Coleoptile length (**F**) and first true leaf length (**G**) of the WT and *AtHFR1* transgenic lines (#498 and #511) under high-B (50 μmol·m^−2^·s^−1^) or low-B (2 μmol·m^−2^·s^−1^) conditions for 7 days. Data are means ± SDs, *n* ≥ 30. Asterisks indicate significant differences in #498 or #511 with Fielder according to ANOVA. ns indicates no significant difference; ** *p* < 0.01. See also Appendix A.

**Figure 2 ijms-23-12057-f002:**
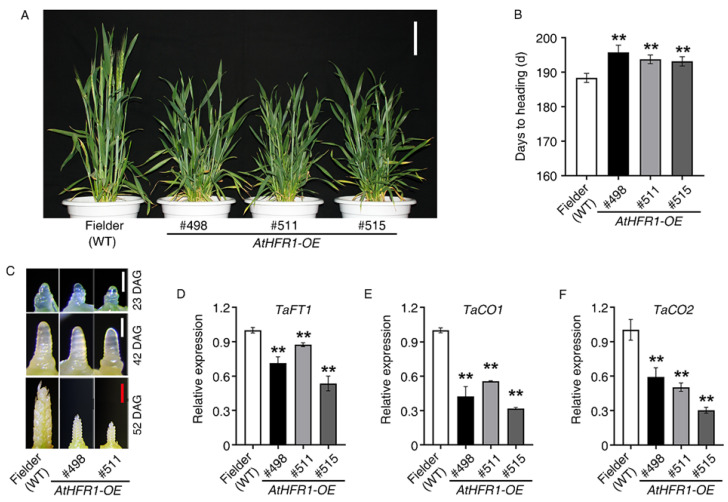
The *AtHFR1* transgene delays heading in common wheat: (**A**) Morphology of the WT (cv. Fielder) and *AtHFR1* transgenic lines (#498, #511, and #515) at heading (181 days after sowing). Bar = 10 cm. (**B**) Heading data for the WT and *AtHFR1* transgenic lines (#498, #511, and #515). Emergence of the first ear from the flag leaf sheath was recognized as the heading date. Data are means ± SDs, *n* ≥ 25. (**C**) Development of the main shoot apex of the WT and *AtHFR1* transgenic lines (#498 and #511). Seedlings were grown under long-day conditions (14 h light/10 h dark; W light 100 μmol·m^−2^·s^−1^, 22 °C) until 23, 42, or 52 DAG (days after germination). White bars = 0.25 mm, red bar = 1 mm. Real-time RT-PCR analysis of *TaFT* (**D**), *TaCO1* (**E**), and *TaCO2* (**F**) in the WT and *AtHFR1* transgenic lines (#498 and #511). Flag leaves were collected at the jointing stage (148 days after sowing). Column shows the mean expression relative to *TaActin* for three biological replicates. Asterisks denote significant differences in #498, #511, or #515 with the WT according to one-way ANOVA (** *p* < 0.01).

**Figure 3 ijms-23-12057-f003:**
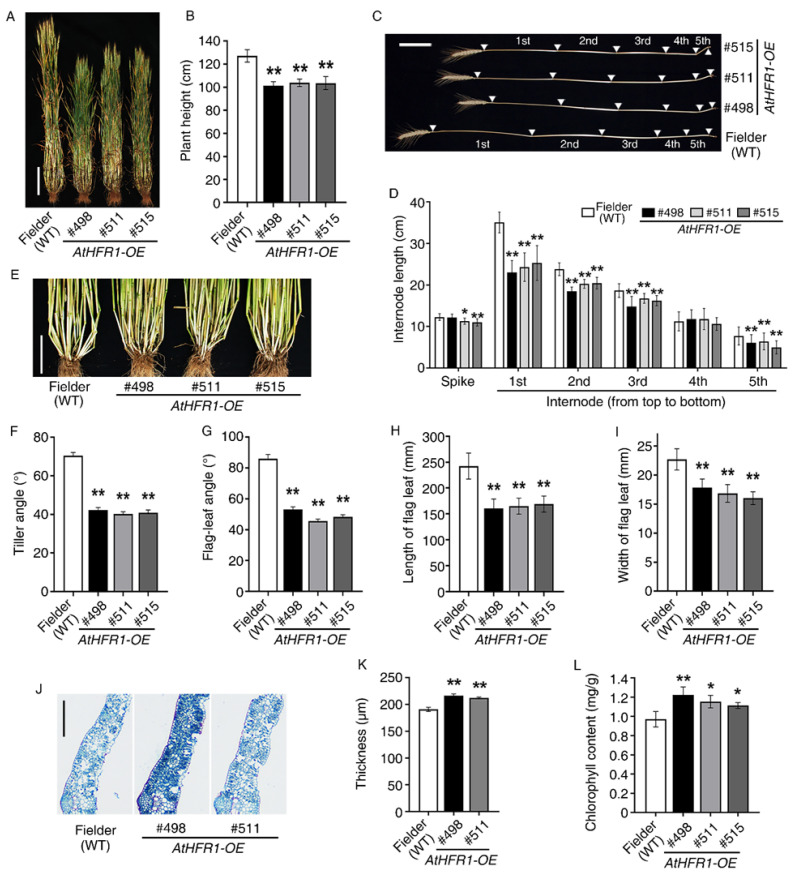
The *AtHFR1* transgene leads to better plant architecture in the field: (**A**) Morphology of the WT (cv. Fielder) and *AtHFR1* transgenic lines (#498, #511, and #515) at the late filling stage (238 days after sowing). Bar = 20 cm. (**B**) Plant height of the WT and *AtHFR1* transgenic lines (#498, #511, and #515). Data are means ± SDs, *n* ≥ 40. Plant internode morphology (**C**) and internode length (**D**) of the WT and *AtHFR1* transgenic lines (#498, #511, and #515). White triangles indicate nodes, and the distances between each pair of nodes are internode lengths (1st to 5th). Bar = 10 cm. Data are means ± SDs, *n* ≥ 35. Tiller morphology (**E**) and tiller angle (**F**) of the WT and *AtHFR1* transgenic lines (#498, #511, and #515) at the late filling stage. Bar = 10 cm. The angle between the tiller and vertical was recorded as the tiller angle. Data are means ± standard errors (SEs), *n* ≥ 50. (**G**) Flag-leaf angle of transgenic lines. The angle between the flag leaf and vertical was recorded as the flag-leaf angle. Data are means ± SEs, *n* ≥ 130. Length (**H**) and width (**I**) of the flag leaf of the WT and *AtHFR1* transgenic lines (#498 and #511). The length between the pulvinus and tip was recorded as the leaf length, and the width at the middle of the leaf was measured as the leaf width. Data are means ± SDs, *n* ≥ 85 (**J**,**K**). Flag leaf thickness of the WT and *AtHFR1* transgenic lines (#498 and #511). The thickness of the flag leaf at the filling stage was measured through slice analysis. Bar = 250 μm. (**L**) Flag leaf chlorophyll contents of the WT and *AtHFR1* transgenic lines (#498, #511, and #515) at the jointing stage (148 days after sowing). Data are means ± SDs, *n* = 5. Asterisks denote significant differences in #498, #511, or #515 with Fielder according to ANOVA (* *p* < 0.05; ** *p* < 0.01).

**Figure 4 ijms-23-12057-f004:**
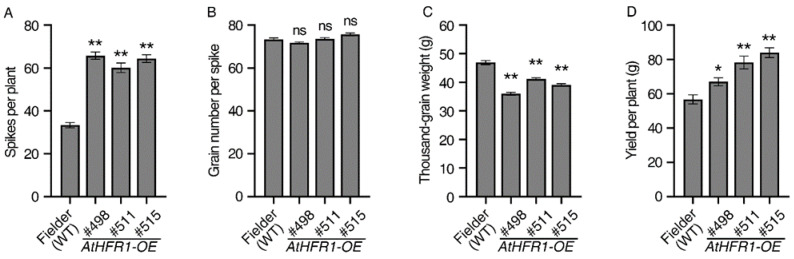
The *AtHFR1* transgene improves grain yield in wheat: Spike number per plant (**A**), grain number per spike (**B**), thousand-gain weight (**C**), and grain yield per plant (**D**) of the WT (cv. Fielder) and *AtHFR1* transgenic lines (#498, #511, and #515). Data are means ± SEs, *n* = 35 (**A**,**B**,**D**), *n* = 15 (**C**). Asterisks denote significant differences in #498, #511, or #515 with the WT according to one-way ANOVA (ns indicates no significant difference; * *p* < 0.05; ** *p* < 0.01). See also Appendix A.

**Figure 5 ijms-23-12057-f005:**
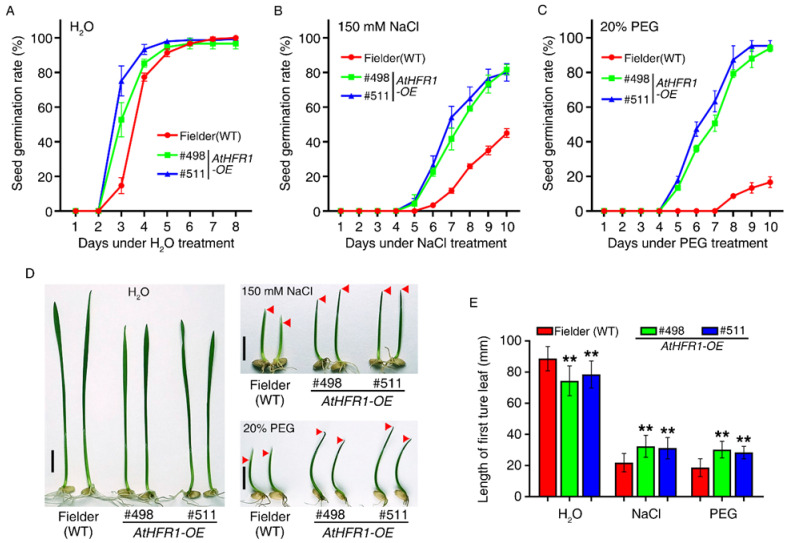
The *AtHFR1* transgene enhances wheat tolerance to osmotic stress during seed germination: Seed germination rates of the WT (cv. Fielder) and *AtHFR1* transgenic lines (#498 and #511) treated with H_2_O (**A**), 150 mM NaCl (**B**), or 20% PEG (**C**). (**D**) Seedling morphology of the WT and *AtHFR1* transgenic lines (#498 and #511) treated with H_2_O, 150 mM NaCl, or 20% PEG for 8 days, showing their first true leaves. (**E**) Quantification of the first true leaf length corresponding to (**D**). Data are means ± SDs, *n* ≥ 40. Asterisks denote significant differences in #498 or #511 with the WT according to ANOVA (** *p* < 0.01).

**Figure 6 ijms-23-12057-f006:**
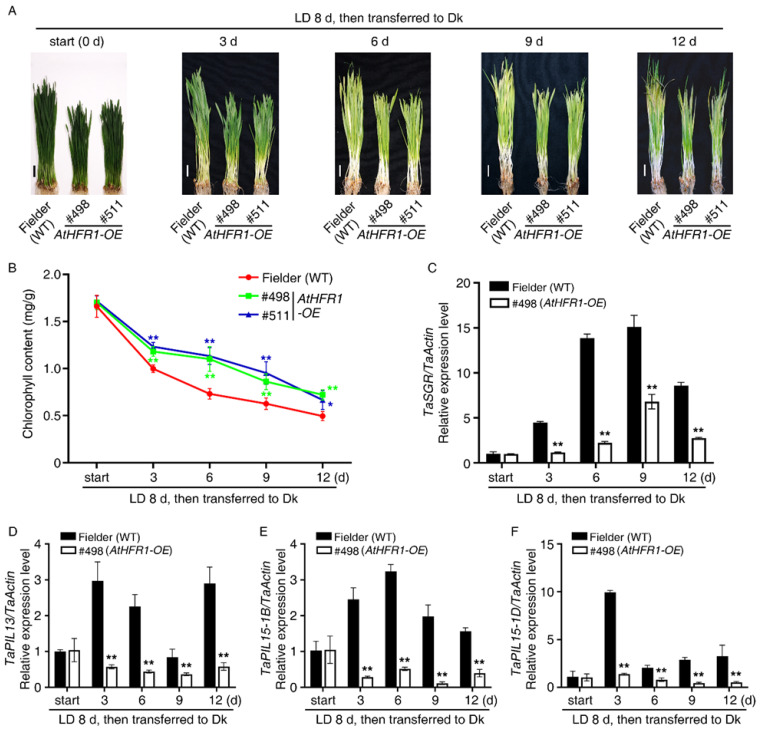
The *AtHFR1* transgene promotes wheat resistance to dark-induced senescence: (**A**) Morphology of the WT (cv. Fielder) and *AtHFR1* transgenic lines (#498 and #511) treated with light starvation. Seedlings were grown under long-day conditions for 8 days, and then transferred to the dark for 0, 3, 6, 9, or 12 days. Bars = 2 cm. (**B**) Chlorophyll contents corresponding to (**A**). Mean and SD (*n* ≥ 5) are shown. Real-time RT-PCR analysis of *TaSGR* (**C**), *TaPIL13* (**D**), *TaPIL15-1B* (**E**), and *TaPIL15-1D* (**F**). Data are means ± SDs, *n* = 3. Asterisks denote significant differences in #498 or #511 with the WT according to ANOVA (* *p* < 0.05, ** *p* < 0.01). See also Appendix A.

## Data Availability

Not applicable.

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
