# Peer review of "HFR1, a bHLH Transcriptional Regulator from Arabidopsis thaliana, Improves Grain Yield, Shade and Osmotic Stress Tolerances in Common Wheat"

_ijms, 2022, doi:10.3390/ijms231912057_

Round 1
Reviewer 1 Report
Dear Author,
I enjoy the paper and I put most of the ratings as "high". Nevertheless there are some things which should be adjusted.
In particular the description in the Material and methods are largely uncompleted.
More information about the transgenic lines should be provided. In the M&M section, variety/varieties used is not mentioned, and more important it is not mention if it is winter or spring wheat. Probably both typologies should be analyzed in order to se possible differences between them.
It is not mentioned if the agronomic traits were measured in replicates. As should do.!
It is not described the NaCl application neither in methods nor in quantity.
It is mentioned that "approximately 215,901,958" but it is not an "approximal" number....
Please change "By 2050, the global population will grow by" into "It is estimated that by 2050 the global population will grow by"
Author Response
Reviewer 1
Dear Author,
I enjoy the paper and I put most of the ratings as "high". Nevertheless, there are some things which should be adjusted.
In particular, the description in the Material and methods are largely uncompleted.
More information about the transgenic lines should be provided. In the M&M section, variety/varieties used is not mentioned, and more important it is not mention if it is winter or spring wheat. Probably both typologies should be analyzed in order to see possible differences between them.
A: Thank you so much. The transgenic background is cv. Fielder, which is a spring type variety, so that it can normally head without strong vernalization treatment. We modified the description in the corresponding parts (Results 2.1 and Materials and Methods 4.1).
It is not mentioned if the agronomic traits were measured in replicates. As should do.!
A: Thanks. In Table S2, S3 in our new version, we listed the agronomic traits of our transgenic lines investigated in another year (2020). Thus, we provided the results of two years (2021 in body content).
It is not described the NaCl application neither in methods nor in quantity.
A: Thanks. We described the method of NaCl application in our new version.
It is mentioned that "approximately 215,901,958" but it is not an "approximal" number....
A: We removed "approximately". Thanks a lot.
Please change "By 2050, the global population will grow by" into "It is estimated that by 2050 the global population will grow by"
A: Thanks. We revised the sentence according to the suggestion.
Reviewer 2 Report
The work entitled “HFR1, a bHLH transcription factor from Arabidopsis thaliana, improves grain yield, shade avoidance, and osmotic stress resistance in common wheat” by Sun, et al. describes the phenotypic characterization of transgenic wheat lines overexpressing the photomorphogenic positive regulator HFR1 from Arabidopsis thaliana. The authors base their work on previous data from other species where the overexpression of positive regulators of photomorphogenesis repress shade avoidance syndrome (SAS) effects leading to increases in yield. Their results suggest that HFR1-OE lines display reduced etiolation responses, increased grain yield, delayed flowering, suppressed dark senescence, produced shorter, sturdier plants and even lead to partial resistance to osmotic stress during germination.
The text is overall well written in a clear and concise manner, their results are well presented and their conclusions are properly placed.
I believe the manuscript has interesting data for wheat breeders aiming for yield increase. Although the manuscript is well structured, I believe it need to clarify some points:
Material and Methods:
Line 390 - the protocol used for wheat transformation is not specified nor cited. This should be clearly stated.
Line 420 - The section 4.3 does not specify the location, coordinates, climate and soil conditions, time of the year, rainfall, and all other relevant information for a field trial.
Line 449 - the method for calculation of relative gene expression by qPCR is not sufficiently described nor cited.
Line 452 and 468 – Please provide the respective gene codes for all genes being analyzed, including TaCO1 and TaCO2.
In the text these points need clarification:
-Figure 1B-C. The authors do not discuss the striking differences between the responses seen for the two independent transgenic lines and they very briefly justify the choice of the lines used in the study. FigS1B shows a non-quantitative method to demonstrate the transgene expression. Are these lines expressing HFR1 at similar levels?
-The claim that HFR1-OE plants display increased lodging resistance is not backed by their results as no specific lodging analysis was performed with these plants. The authors base their conclusion solely on morphological parameters such as height and tiller angle, but these are insufficient for claiming that the plants would behave better in the field.
-Line 174 and Figure 2D-E. The gene expression data does not consider that these genes are highly responsive to circadian oscillation see Shaw et al., 2020 https://doi.org/10.1371/journal.pgen.1008812) . Either the authors should explicit the ZT time of sampling or some other manner to confirm that these samples can be compared.
-Line 356. As HFR1 does not directly bind DNA (line 72), it is not considered a bona fide transcription factor, and more like a transcriptional regulator. Therefore, the authors should avoid calling HFR1 a transcription factor.
Author Response
Reviewer 2
The work entitled “HFR1, a bHLH transcription factor from Arabidopsis thaliana, improves grain yield, shade avoidance, and osmotic stress resistance in common wheat” by Sun, et al. describes the phenotypic characterization of transgenic wheat lines overexpressing the photomorphogenic positive regulator HFR1 from Arabidopsis thaliana. The authors base their work on previous data from other species where the overexpression of positive regulators of photomorphogenesis repress shade avoidance syndrome (SAS) effects leading to increases in yield. Their results suggest that HFR1-OE lines display reduced etiolation responses, increased grain yield, delayed flowering, suppressed dark senescence, produced shorter, sturdier plants and even lead to partial resistance to osmotic stress during germination.
The text is overall well written in a clear and concise manner, their results are well presented and their conclusions are properly placed.
I believe the manuscript has interesting data for wheat breeders aiming for yield increase. Although the manuscript is well structured, I believe it need to clarify some points:
Material and Methods:
Line 390 - the protocol used for wheat transformation is not specified nor cited. This should be clearly stated.
A: In the new version, we added the wheat transformation method.
Line 420 - The section 4.3 does not specify the location, coordinates, climate and soil conditions, time of the year, rainfall, and all other relevant information for a field trial.
A: Thanks for your suggestion. We marked the location, coordinates, climate, and soil conditions in the new version.
Line 449 -the method for calculation of relative gene expression by qPCR is not sufficiently described nor cited.
A: In the new version, we provided the method for calculation of relative gene expression by qPCR.
Line 452 and 468 – Please provide the respective gene codes for all genes being analyzed, including TaCO1 and TaCO2.
A: We did in the new version. Thanks a lot.
In the text these points need clarification:
-Figure 1B-C. The authors do not discuss the striking differences between the responses seen for the two independent transgenic lines and they very briefly justify the choice of the lines used in the study. FigS1B shows a non-quantitative method to demonstrate the transgene expression. Are these lines expressing HFR1 at similar levels?
A: We did qRT-PCR, we found that the relative expression levels of the transgenic AtHFR1 in Line #511 and #515 were 1.12- and 1.26-fold higher than that in Line #498 (Figure S2 in the new version).
-The claim that HFR1-OE plants display increased lodging resistance is not backed by their results as no specific lodging analysis was performed with these plants. The authors base their conclusion solely on morphological parameters such as height and tiller angle, but these are insufficient for claiming that the plants would behave better in the field.
A: Thank you for your suggestions. We changed “HFR1-OE plants display increased lodging resistance” to “HFR1-OE plants display increased shade tolerance”.
-Line 174 and Figure 2D-E. The gene expression data does not consider that these genes are highly responsive to circadian oscillation see Shaw et al., 2020 https://doi.org/10.1371/journal.pgen.1008812). Either the authors should explicit the ZT time of sampling or some other manner to confirm that these samples can be compared.
A: We checked the relative expression levels of TaFT1, TaCO1 and TaCO2 in #498, #511, or #515 and the WT again using seedlings of LD-to-Dk transition for 3 days, and also confirmed the AtHFR1 transgene repressed TaFT1, TaCO1 and TaCO2 expression. See the new Figure S3.
-Line 356. As HFR1 does not directly bind DNA (line 72), it is not considered a bona fide transcription factor, and more like a transcriptional regulator. Therefore, the authors should avoid calling HFR1 a transcription factor.
A: Thanks for your suggestion. We used “transcriptional regulator” instead of “transcription factor” to describe HFR1 in the new version.
